# Agent Modelling under Partial Observability for Deep Reinforcement Learning

**Georgios Papoudakis**    **Filippos Christianos**    **Stefano V. Albrecht**

School of Informatics
University of Edinburgh
`{g.papoudakis, f.christianos, s.albrecht}@ed.ac.uk`

## Abstract

Modelling the behaviours of other agents is essential for understanding how agents interact and making effective decisions. Existing methods for agent modelling commonly assume knowledge of the local observations and chosen actions of the modelled agents during execution. To eliminate this assumption, we extract representations from the local information of the controlled agent using encoder-decoder architectures. Using the observations and actions of the modelled agents during training, our models learn to extract representations about the modelled agents conditioned only on the local observations of the controlled agent. The representations are used to augment the controlled agent's decision policy which is trained via deep reinforcement learning; thus, during execution, the policy does not require access to other agents' information. We provide a comprehensive evaluation and ablations studies in cooperative, competitive and mixed multi-agent environments, showing that our method achieves higher returns than baseline methods which do not use the learned representations.

## 1 Introduction

An important aspect of autonomous decision-making agents is the ability to reason about the unknown intentions and behaviours of other agents. Much research has been devoted to this *agent modelling* problem [Albrecht and Stone, 2018], with recent works focused on learning informative representations about another agent's policy using deep learning architectures for agent modelling and reinforcement learning (RL) [He et al., 2016, Raileanu et al., 2018, Grover et al., 2018, Rabinowitz et al., 2018, Zintgraf et al., 2021].

A common assumption in existing methods is that the modelling agent has access to the local trajectory of the modelled agents during execution [Albrecht and Stone, 2018], which may include their local observations of the environment state and their past actions. While it is certainly desirable to be able to observe the agents' local contexts in order to reason about their past and future decisions, in practice such an assumption may be too restrictive. Agents may only have a limited view of their surroundings, communication with other agents may be infeasible or unreliable [Stone et al., 2010], and knowledge of the perception system of other agents may be unavailable [Gmytrasiewicz and Doshi, 2005]. In such cases, an agent must reason with only locally available information.

We consider the following question: *Can effective agent modelling be achieved using only the locally available information of the controlled agent during execution?* A strength of deep learning techniques is their ability to identify informative features in data. Here, we use deep learning techniques to extract informative features about the trajectory of the modelled agent from a stream of local observations for the purpose of agent modelling. Specifically, we consider a multi-agent setting in which we control one agent which must learn to interact with a set of other agents. We assume a set of possible policies for the non-learning agents and that these policies are fixed.

35th Conference on Neural Information Processing Systems (NeurIPS 2021).

We propose *Local Information Agent Modelling* (LIAM)[1], which can be seen as the general idea of learning the relationship between the trajectory of the controlled agent and the trajectory of the modelled agent. In this work we propose one instantiation of this idea: an encoder-decoder agent modelling method that can extract a compact yet informative representation of the modelled agents given only the local information of the controlled agent (its local state observations, and past actions). The model is trained to replicate the observations and actions of the modelled agents from the local information only. During training, the modelled agent's observations and actions are utilised as reconstruction targets for the decoder; after training, only the encoding component is retained which generates representations using local observations of the controlled agent. The learned representation conditions the policy of the controlled agent in addition to its local observation, and the policy and model are optimised concurrently during the RL learning process.

We evaluate LIAM in three different multi-agent environments: double speaker-listener [Mordatch and Abbeel, 2017, Lowe et al., 2017], level-based foraging (LBF) [Albrecht and Stone, 2017, Papoudakis et al., 2021], and a modified version of predator-prey proposed by [Böhmer et al., 2020]. Our results support the idea that effective agent modelling can be achieved using only local information during execution: the same RL algorithm generally achieved higher average returns when combined with representations generated by LIAM than without, and in some cases the average returns are comparable to those achieved by an ideal baseline which has access to the modelled agent's trajectory during execution. We also provide detailed evaluations of the learned encoder and decoder of LIAM as well as comparison with different instantiations of LIAM.

## 2 Related Work

**Learning Agent Models:** We are interested in agent modelling methods that use neural networks to learn representations of the other agents. He et al. [2016] proposed a method which learns a modelling network to reconstruct the modelled agent's actions given its observations. Raileanu et al. [2018] developed an algorithm for learning to infer an agent's intentions using the policy of the controlled agent. Grover et al. [2018] proposed an encoder-decoder method for modelling the agent's policy. The encoder learns a point-based representation of different agent trajectories, and the decoder learns to reconstruct the modelled agent's policy. Rabinowitz et al. [2018] proposed the Theory of mind Network (TomNet), which learns embedding-based representations of modelled agents for meta-learning. Tacchetti et al. [2019] proposed relational forward models to model agents using graph neural networks. [Zintgraf et al., 2021] uses a VAE for agent modelling for fully-observable tasks. All these aforementioned works either assume that the controlled agent has direct access to the trajectories of the modelled agent during execution or that the evaluation environments are fully-observable. Xie et al. [2020] learned latent representations from local information to influence the modelled agents, however, in contrast to our work, they did not utilise the modelled agent's trajectories. Finally, agent modelling from local information has been researched under the I-POMDP model [Gmytrasiewicz and Doshi, 2005] and in the Poker domain research. In contrast to our work, I-POMDPs utilise recursive reasoning [Albrecht and Stone, 2018] which assumes knowledge of the observation models of the modelled agents (which is unavailable in our setting). In the Poker domain, Johanson et al. [2008], Bard et al. [2013] created a mixture-of-expert counter-strategies during training against several type of opponent policies, while during execution they cast the selection of the best counter-strategy against a specific opponent type as a bandit problem. The main difference between LIAM and these works is that the latter adapt (to select the best strategy) over a number of hands (we consider that a hand is equivalent to an episode) against each opponent. In contrast, in our work we use a single episode for adaptation.

**Representation Learning in Reinforcement Learning:** Another related topic which has received significant attention is representation learning in RL. Using unsupervised learning techniques to learn low-dimensional representations of the environment state has led to significant improvements both in the returns as well as the sample efficiency of the RL. Recent works on representation learning focus on learning world models and use them to train the RL algorithm [Ha and Schmidhuber, 2018, Hafner et al., 2020] or learning state representations for improving the sample efficiency of RL algorithms using reconstruction-based [Corneil et al., 2018, Igl et al., 2018, Kurutach et al., 2018, Gregor et al., 2019, Gelada et al., 2019] or non-reconstruction-based methods [Laskin et al., 2020, Zhang et al., 2020], using contrastive learning or bi-simulation metrics. Works on meta-RL and transfer learning

---

[1]We provide an implementation of LIAM in `https://github.com/uoe-agents/LIAM`

focus on learning representations over tasks and use them to solve new and previously unseen tasks [Doshi-Velez and Konidaris, 2016, Hausman et al., 2018, Zhang et al., 2018, Rakelly et al., 2019, Zintgraf et al., 2019]. In contrast to these work, LIAM focuses on learning representations about the relationship between the trajectories of the controlled agent and of the modelled agent.

**Multi-agent Reinforcement Learning (MARL):** MARL algorithms are designed to train multiple agents to solve tasks in a shared environment [Hernandez-Leal et al., 2017, Papoudakis et al., 2019]. At the beginning of the training, a number of agents are untrained and typically initialised with random policies. A common paradigm in MARL is Centralised Training with Decentralised Execution (CTDE), in which the trajectories of all agents are utilised during training, while during execution each agent conditions its individual policy only on its local trajectory. The information of all agents can be utilised during training for various reasons, such as computing a joint value function [Lowe et al., 2017, Foerster et al., 2018, Sunehag et al., 2018, Rashid et al., 2018], or generating intrinsic rewards [Jaques et al., 2019]. During execution, only the local information of each agent is used for selecting actions in the environment. Our work relates to CTDE in that we train LIAM in a centralised fashion using the trajectories of all the existing agents in the environment. However, during execution, the learned model uses only the local trajectory of the controlled agent.

## 3 Approach

### 3.1 Problem Formulation

We control a single agent which must learn to interact with other agents that use one of a fixed number of policies. We model the problem as a Partially-Observable Stochastic Game (POSG) [Shapley, 1953, Hansen et al., 2004] which consists of $N$ agents $\mathbb{I} = \{1, 2, ..., N\}$, a state space $\mathcal{S}$, the joint action space $\mathcal{A} = \mathcal{A}^1 \times ... \times \mathcal{A}^N$, a transition function $P : \mathcal{S} \times \mathcal{A} \times \mathcal{S} \rightarrow [0, 1]$ specifying the transition probabilities between states given a joint action, and for each agent $i \in \mathbb{I}$ a reward function $r^i : \mathcal{S} \times \mathcal{A} \times \mathcal{S} \rightarrow \mathbb{R}$. We consider that each agent $i$ has access to its observation $o^i \in \mathcal{O}^i$, where $\mathcal{O}^i$ is the observation set of agent $i$. The observation function $\Omega^i : \mathcal{S} \times \mathcal{A} \times \mathcal{O}^i \rightarrow [0, 1]$ defines a probability distribution over the possible next observations of agent $i$ given the previous state and the joint action of all agents.

We denote the agent under our control by $1$, and the modelled agents by $-1$ where for notational convenience we will treat the modelled agents as a single "combined" agent with joint observations $o^{-1}$ and actions $a^{-1}$. We assume a set of fixed policies, $\Pi = \{\pi^{-1,k} | k = 1, ..., K\}$, which may be defined manually (heuristic) or pretrained using RL. Each fixed policy determines the modelled agent's actions as a mapping $\pi^{-1,k}(o^{-1})$ from the modelled agent's local observation $o^{-1}$ to a distribution over actions $a^{-1}$. Our goal is to find a policy $\pi_\theta$ parameterised by $\theta$ for agent 1 which maximises the average return against the fixed policies from the training set $\Pi$, assuming that each fixed policy is initially equally probable and fixed during an episode:

$$\arg \max_\theta \mathbb{E}_{\pi_\theta, \pi^{-1,k} \sim \mathcal{U}(\Pi)} \left[ \sum_{t=0}^{H-1} \gamma^t r^1_{t+1} \right] \tag{1}$$

where $r^1_{t+1}$ is the reward received by agent 1 at time $t + 1$ after performing the action $a^1_t$, $H$ is the episode length (horizon), and $\gamma \in (0, 1)$ is a discount factor. It is also important to note that neither during training nor during execution the controlled agent has access to the identity $k$ of the policy that is used by the modelled agent.

### 3.2 Local Information Agent Modelling

We aim to learn the relationship between the trajectory of the controlled agent and the trajectory of the modelled agent. We denote by $\tau^{-1} = \{o_t^{-1}, a_t^{-1}\}_{t=0}^{t=H}$ the trajectory of the modelled agent where $o_t^{-1}$ and $a_t^{-1}$ are the modelled agent's observation and action at time step $t$ in the trajectory, up to horizon $H$. These trajectories are generated from the fixed policies in $\Pi$. We assume the existence of some latent variables (or *embeddings*) in the space $\mathcal{Z}$, and at each time step $t$ the latent variables $z_t$ contain information both about the fixed policy that is used by the modelled agent as well as the dynamics of the environment as perceived by the modelled agent. To learn the relationship between the modelled agent's trajectory and the latent variables we can use a parametric decoder. The decoder

is denoted as $f_{\boldsymbol{u}} : \mathcal{Z} \to \tau^{-1}$ and is the model that decodes the latent variables to the trajectory of the modelled agent.

The last step to learn the relationship between the local and the modelled agent's trajectories is to use a recurrent encoding model, that we denote as $f_{\boldsymbol{w}} : \tau^1 \to \mathcal{Z}$, with parameters $\boldsymbol{w}$, to learn the relationship between the local trajectory of the controlled agent and the latent variables. Specifically, we learn the function that relates the modelled agent's trajectory to the latent variables with an encoder that only depends on local information of the controlled agent. Since during execution only the encoder is required to generate the latent variables of the modelled agent, this approach removes the assumption that access to the modelled agent's observations and actions is available during execution.

At each time step $t$, the recurrent encoder network generates an embedding $z_t$, which is conditioned on the information of the agent under control $(o^1_{1:t}, a^1_{1:t-1})$, until time step $t$. At each time step $t$ the parametric decoder learns to reconstruct the modelled agent's observation and action ($o^{-1}_t$ and $a^{-1}_t$) conditioned on the embedding $z_t$. Therefore, the decoder consists of a fully-connected feed-forward network with two output heads; the observation reconstruction head $f^o_{\boldsymbol{u}}$, and the policy reconstruction head $f^\pi_{\boldsymbol{u}}$ (see Figure 1). In each time step $t$, the decoder receives as input embedding $z_t$ and the observation reconstruction head reconstructs the modelled agent's observation $o^{-1}_t$, while the action reconstruction head outputs a categorical distribution over the modelled agent's action $a^{-1}_t$. We observe that the output dimensions of the two decoder's reconstruction heads grow linearly

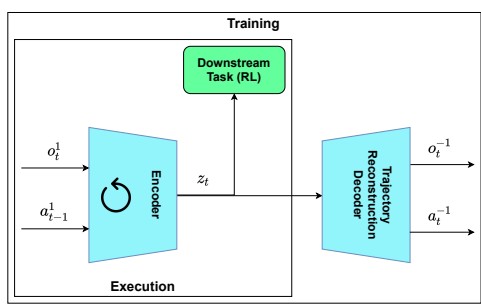

Figure 1: Diagram of LIAM architecture. The two solid-line rectangles show the components of LIAM that are used during training and during execution respectively.

with respect to the number of the agents in the environment. We refer to this method as **LIAM** (**L**ocal **I**nformation **A**gent **M**odelling). LIAM uses the information of both the controlled agent and the modelled agent during training, but during execution only the information of the controlled agent is used. The encoder-decoder loss is defined as:

$$\mathcal{L}_{ED} = \frac{1}{H} \sum_{t=1}^{H} [(f^o_{\boldsymbol{u}}(z_t) - o^{-1}_t)^2 - \log f^\pi_{\boldsymbol{u}}(a^{-1}_t | z_t)] \quad \text{where} \ z_t = f_{\boldsymbol{w}}(o^1_{:t}, a^1_{:t-1}) \quad (2)$$

### 3.3 Reinforcement Learning Training

The latent variables $z$ augmented with the controlled agent's observation can be used to condition the RL optimised policy. Consider the augmented space $\mathcal{O}^1_{aug} = \mathcal{O}^1 \times \mathcal{Z}$, where $\mathcal{O}^1$ is the original observation space of the controlled agent in the POSG, and $\mathcal{Z}$ is the representation space about the agent's models. The advantage of learning the policy on $\mathcal{O}^1_{aug}$ compared to $\mathcal{O}^1$ is that the policy can specialise for different $z \in \mathcal{Z}$. In our experiments we optimised the policy of the controlled agent using A2C, however, other RL algorithms could be used in its place. The input to the actor and critic are the local observation and the generated representation. We do not back-propagate the gradient from the actor-critic loss to the parameters of the encoder. We use different learning rates for optimising the parameters of the networks of RL and the encoder-decoder. We empirically observed that LIAM exhibits high stability during learning, allowing us to use larger learning rate compared to RL. Additionally, we subtract the policy entropy from the policy gradient loss to encourage exploration [Mnih et al., 2016]. Given a batch $B$ of trajectories, the objective of A2C is:

$$\mathcal{L}_{A2C} = \mathbb{E}_{(o_t, a_t, r_{t+1}, o_{t+1}) \sim B} [\frac{1}{2} (r^1_{t+1} + \gamma V_{\boldsymbol{\phi}}(o^1_{t+1}, z_{t+1}) - V_{\boldsymbol{\phi}}(o^1_t, z_t))^2$$
$$- \hat{A} \log \pi_\theta(a^1_t | o^1_t, z_t) - \beta H(\pi_\theta(a^1_t | o^1_t, z_t))] \quad (3)$$

The pseudocode of LIAM is given in Appendix A and the implementation details in Appendix D. Intuitively, at the beginning of each episode, LIAM starts with uninformative embeddings "average" over the possible agent trajectories. At each time step, the controlled agent interacts with the environment and the modelled agent, and updates the embeddings based on the local trajectory that it perceives.

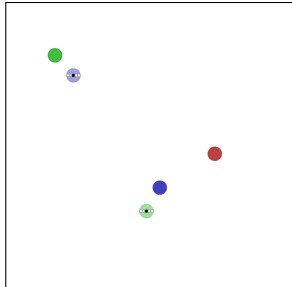 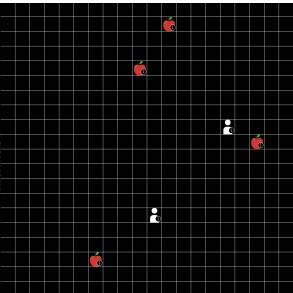 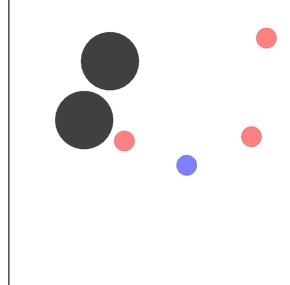

Figure 2: The three evaluation environments. Double speaker-listener (left), level-based foraging (middle), predator prey (right).

# 4 Experiments

## 4.1 Multi-Agent Environments

We evaluate the proposed method in three multi-agent environments (one cooperative, one mixed, one competitive): double speaker-listener [Mordatch and Abbeel, 2017], level-based foraging [Albrecht and Stone, 2017, Papoudakis et al., 2021], and a version of predator-prey proposed by [Böhmer et al., 2020]. For each environment, we create ten different policies which are used for training (set $\Pi$). More details about the process of generating the fixed policies are presented in Appendix B.

**Double speaker-listener (DSL):** The environment consists of two agents and three designated landmarks. At the start of each episode, the agents and landmarks are generated in random positions, and are randomly assigned one of three possible colours - red, green, or blue. The task of each agent is to navigate to the landmark that has the same colour. However, each agent cannot perceive its colour. The colour of each agent can only be observed by the other agents. Each agent must learn both to communicate a five-dimensional message to the other agent as well as navigate to the correct landmark. The controlled agent's observation includes the relative positions of all landmarks and the other agents as well as the communication message from the previous time step and the colour of the other agent. The cooperative reward at each time step is the negative average Euclidean distance between two agents and their corresponding correct landmark.

**Level-based foraging (LBF):** The environment is a $20 \times 20$ grid-world, consisting of two agents and four food locations. The agents and the foods are assigned random levels and positions at the beginning of an episode. The goal is for the agents to collect all foods. Agents can either move in one of the four directions or attempt to pick up a food. A group of one or more agents successfully pick a food if the agents are positioned in the adjacent cells to the food and if the sum of the agents' levels is at least as high as the food's level. The controlled agent has to learn to cooperate to load foods with a high level and at the same time act greedily for foods that have lower levels. The environment has sparse rewards, representing the contribution of the agent in the gathering all foods in the environment. For example, if the agent receives a food with level 2, and there are another three foods with levels 1, 2 and 3 respectively, the reward of the agent is $2/(1 + 2 + 2 + 3) = 0.25$. The maximum cumulative reward that all agents can achieve is normalised to 1. The environment is partially-observable, and the controlled agent perceives other agents and foods that are located up to four grid cells in every direction. However, the modelled agents can observe the full environment state. The episode terminates when all available foods have been loaded or after 50 time steps.

**Predator Prey (PP):** The environment consists of four agents and two large obstacles. Three of the agents are predators and the other agent is the prey. Each agent has five navigation actions. If only one of the predators captures the prey then the predators receive reward $-1$ and the prey reward 1. If two or more predators capture the prey, then the predators receive reward 1 and the prey reward $-1$. In the experiments, we control the prey, while the predators sample their policies from the set of fixed policies. The environment is partially-observable where the prey can only observe other agents and obstacles that are within its receptive field, while the predators observe all agents and obstacles in the environment. The episode terminates after 50 time steps.

## 4.2 Baselines

Our problem of learning to adapt to different fixed policies can be viewed either as an agent modelling problem, or as a single-agent task-adaptation RL problem in which we control one agent which has to learn to adapt to multiple tasks, where each one of the fixed policies defines a different task. Thus, task adaptation algorithms can be used as baselines to address our problem. We compare our method against five baselines, two of which are indicative of the upper and the lower performance of LIAM when it is evaluated against the policies from $\Pi$. All baselines are trained using the A2C algorithm.

**Full Information Agent Model (FIAM):** This baseline is indicative of the upper performance of LIAM when it is evaluated against the fixed policies from the set $\Pi$. FIAM utilises the trajectories of the modelled agent during training as well as during execution and it is trained similarly to LIAM. However, the encoder is conditioned on the actual trajectory of the modelled agent, and therefore requires access to the modelled agent's trajectory during execution. Since FIAM has access to more information than LIAM during execution, we intuitively expect to achieve higher returns compared to LIAM.

**No Agent Model (NAM):** This baseline does not use an explicit agent model, and it is indicative of the lower performance of LIAM when it is evaluated against the fixed policies from the set $\Pi$. It does, however, use a recurrent policy network which receives as input the observation and action of the controlled agent. By using a recurrent network we can compare returns achieved by implicit agent modelling (that is done by the hidden states of the recurrent network) with the returns achieved by explicit agent modelling that is done by LIAM. NAM is similar to the single-agent task-adaptation algorithm $RL^2$ [Wang et al., 2016, Duan et al., 2016]. In contrast to the original implementation of $RL^2$, we reset the hidden state of the recurrent network at the beginning of each episode, and we do not use the agent's reward as input to the policy to ensure consistency among the tested algorithms. Since NAM does not have access to the modelled agent's trajectory during training, we intuitively expect to perform worse than LIAM in terms of achieved returns.

**VariBad [Zintgraf et al., 2019]:** This baseline trains a VAE to learn latent representations of the observation and reward functions, as they are perceived by the controlled agent, conditioned on the observation, action, reward triplet of the controlled agent. We include VariBad for two reasons: it is an algorithm that learns to adapt to different tasks, and also it uses a recurrent encoder similarly to LIAM, but does not utilise the trajectory of the modelled agent. As a result, we can observe whether utilising the modelled agent's trajectory during training, as is done in LIAM, results in higher evaluation returns. To ensure consistency among the tested algorithms, we do not use the reward as input in the encoder of VariBad.

**Classification-Based Agent Modelling (CBAM):** This baseline learns to reconstruct the identity of the policy that is used by the modelled agent. The policy of the controlled agent is conditioned on the reconstructed identity concatenated with the local observation of the controlled agent. CBAM consists of a recurrent network that receives as input the observation-action sequence of the controlled agent. The output has a softmax activation function, and we train it to maximise the log-likelihood of the identities of the policies that are used by the modelled agent. CBAM uses the fixed policy identities during training (that LIAM does not), but does not use the trajectories of the modelled agent (that LIAM does).

**Constrastive Agent Representation Learning (CARL):** Finally, we evaluate a non-reconstruction baseline based on contrastive learning [Oord et al., 2018]. CARL utilises the modelled agent's trajectories during training but during execution only the trajectories of the controlled agent are used. Details of the implementation of this baseline are included in Appendix C. This baseline was included because recent works [Laskin et al., 2020, Zhang et al., 2020] in state representation found that non-reconstruction methods tend to perform better than reconstruction methods, especially in domains that have pixel-based observations.

## 4.3 Evaluation of Returns

Figure 3 shows the average evaluation returns of all methods during training in the three multi-agent environments. The returns of every evaluated method are averaged over five runs with different initial seeds, and the shadowed part represents a $95\%$ confidence interval (two standard errors of the mean). We evaluate the policy, learned by each method, every 1000 training episodes for 100 episodes. During the evaluation, the agent follows the stochastic policy that is outputted from the

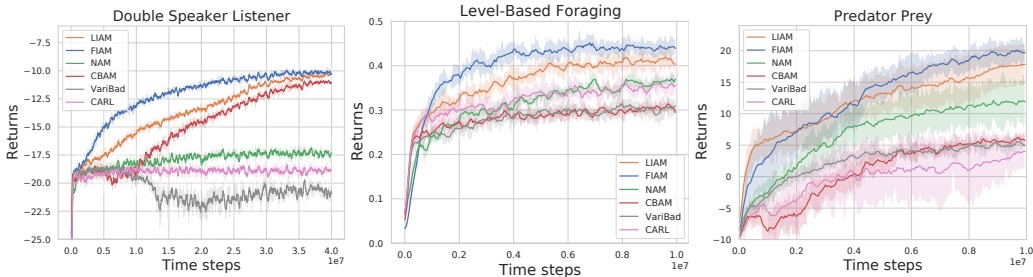

Figure 3: Episodic evaluation returns and 95% confidence interval of the six evaluated methods during training, against policies from Π.

policy network. We found that sampling the action from the policy distribution leads to significantly higher returns compared to following the greedy policy.

First of all, we observe that in all three evaluation environment the returns of LIAM are closer to the upper baseline (FIAM) than to the lower baseline (NAM). The difference in the returns between LIAM and VariBad can be attributed to the fact that LIAM utilises the modelled agent's trajectories during training, leading to better representations for RL. VariBad only learns how the current observation of the controlled agent is related to the next observation and reward, while LIAM learns how the local observation is related to the trajectory of the modelled agent. CARL performs worse compared to LIAM because the generated embeddings contain less information compared to the embeddings generated from LIAM. For example, in the level-based foraging environment, a specific time step $t$ the control agent observes that the modelled agent is at a specific cell. The observation of the modelled agent also contains this type of information, and as a result, the contrastive task can easily relate that these two embeddings are the two parts of a positive pair. Therefore the representation is trained to only encode such information. On the other hand, LIAM encodes into $z$ all information that has been gathered so far in the interaction to be able to reconstruct the modelled agent's trajectory. We observe that LIAM achieves higher return than CBAM because identifying the agent identity is not always an informative representation. LIAM utilises the modelled agent's trajectory during training and learns how the local trajectory corresponds to the trajectory of the modelled agent. In the double speaker-listener environment, CBAM's returns are higher compared to the rest of the baselines and close to the returns of LIAM, because the different fixed policies are discretely different (different communication messages correspond to different colours) and the classifier learns to accurately identify the fixed policy identity. However, in the other environments the classifier is unable to separate different fixed policies, which results in lower average returns.

## 4.4 Model Evaluation

After comparing LIAM against five baselines with respect to the average evaluation returns, we now evaluate the encoder and the decoder of LIAM. First, we visualise the embedding space of LIAM and we evaluate how fast the encoder learns to generate informative embeddings. Finally, we evaluate whether the decoder of LIAM learns to accurately reconstruct the modelled agent's actions.

We analyse the embeddings learned by LIAM's encoder. Figure 4a presents the two-dimensional projection, using the t-SNE method [Van der Maaten and Hinton, 2008], of the generated embeddings at the 20th time step of the episode for all policies in Π in the double speaker-listener environment, where each colour represents a different policy in the set Π. We observe that clusters of similar colours are generated, indicating the interactions with the same fixed policy for the modelled agent result in representations that are close to each other. However, we observe that each colour appears in more than one cluster. We speculate that different clusters can represent the different modalities that exist in the trajectory of the modelled agent, such as the colour of the controlled agent and the message that is communicated from the modelled agent to the controlled agent.

Figure 4b presents how the action reconstruction accuracy, and the accuracy of the controlled agent identifying its own colour (which is not observed by the controlled agent) in the double speaker-listener environment are changing through time. The colour of the controlled agent is represented as three-dimensional one-hot vector in the observation of the modelled agent, and the controlled agent does not have direct access to this information during their interaction. At each time step $t$

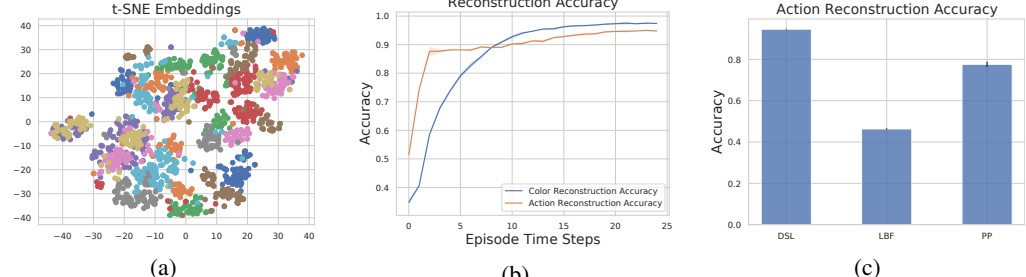

(a)        (b)        (c)

Figure 4: (a) t-SNE projection of the embeddings in the double speaker-listener environment, where different colours represent different fixed policies and different points represent different episodes. (b) Modelled agent's actions and controlled agent's colours reconstruction accuracy with respect to the episode time steps. (c) Reconstruction accuracy of the modelled agent's actions at the 20th time step of the episode in the three evaluation environments (the confidence interval is not visible in the double speaker-listener bar plot).

we use the embedding $z_t$ to reconstruct the observation and the action of the modelled agent. In the reconstructed observation we find the three dimensional vector that corresponds to the colour of the controlled agent, and we consider that the identified colour corresponds to the element of the three-dimensional vector that is closer to 1. Then, we compute the percentage of the times where the reconstructed colour matches the colour of that is truly observed by the modelled agent. LIAM learns to identify the modelled agent's action as well as the colour of the controlled agent with accuracy greater than 90% after 10 time steps.

The decoder of the controlled agent receives as input an embedding at each time step, and predicts the observations and actions of the modelled agent. We evaluate the decoder based on action prediction accuracy. At the 20th time step of the episode, we use the generated embeddings and the decoder to reconstruct the modelled agent's actions. Figure 4c presents the average action prediction accuracy and the 95% confidence interval in the three evaluation environments. The reduced action prediction accuracy in level-based foraging and the predator-prey compared to double speaker-listener can be explained by the fact that the controlled agent does not always observe the modelled agent, and as a result it is harder to predict their actions.

## 4.5   Ablation Study

In this section we perform an ablation study to justify the design choices behind the architecture of LIAM. We design and evaluate six different ablations of LIAM in the double speaker-listener environment. First, we evaluate two ablated versions of LIAM's decoder: LIAM-No-Act-Recon which reconstructs only the observations of the modelled agent, LIAM-No-Obs-Recon which reconstructs only the actions of the modelled agent. Then, we evaluate how different data inputs in the encoder of LIAM affect the returns in the double speaker-listener environment. We consider the model LIAM-No-Act, where in the encoder of LIAM we only input the observations of the controlled agent, and the model LIAM-No-Obs, where in the encoder of LIAM we only input the actions of the controlled agent. We also consider an encoder-decoder model that does not utilise the trajectory of the modelled agent during training. We call

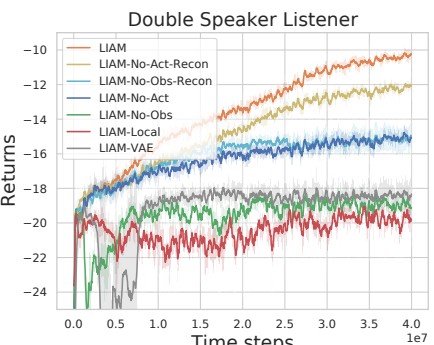

Figure 5: Average returns comparison of LIAM against six ablated versions of LIAM in the double speaker-listener environment.

this model LIAM-Local, and at each time step $t$ the decoder learns to reconstruct the next observation $o_{t+1}^1$ and the action $a_t^1$ conditioned on the embedding $z_t$. Additionally, we evaluate a variational encoder-decoder model [Kingma and Welling, 2014] that performs inference to a Hidden Markov Model (HMM), and we refer to it as LIAM-VAE. LIAM-VAE learns a prior $(p(z_t|z_{t-1}))$ that express

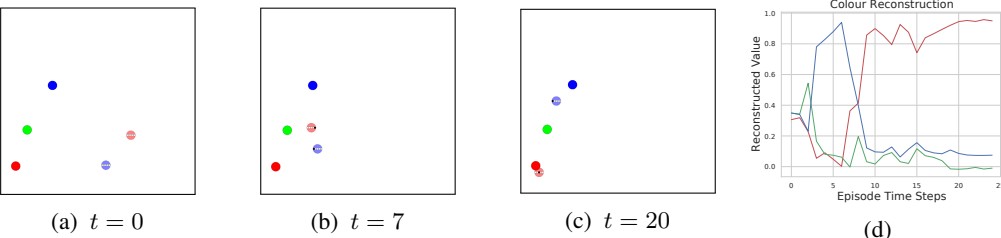

| (a) $t = 0$ | (b) $t = 7$ | (c) $t = 20$ | (d) |

Figure 6: (a, b, c) Snapshots of one episode of the double speaker-listener environment at time steps 0, 7 and 20 respectively. (d) Decoder's reconstruction values of the colour of the controlled agent.

the temporal relationship of the latent distribution, similarly to Chung et al. [2015]. We discuss the implementation of this model in Appendix C.

Figure 5 presents the average returns of LIAM and the aforementioned ablation baselines in the double speaker-listener environment. First, we observe that non-reconstructing either the observation or the action of the controlled agent leads to significantly lower returns than the original implementation of LIAM. We also observe that the lack of either the observation or the action of the controlled agent from the encoder of LIAM results in significant lower returns compared to full LIAM. The reason is that the model observes how the modelled agent reacts (by observing the relative position of the modelled agent that is included in the controlled agent's observation) to different communication messages (the communication message is one of the actions of the controlled agent) that the controlled agent outputs, and learns to generate informative embeddings (see also Section 4.6). LIAM-Local achieves the lowest returns compared to the other baselines. The main reason behind this result, is that LIAM-Local does not learn the relationship between the local and the modelled agent's trajectory. Finally, we believe that the main reason that LIAM-VAE performs worse than LIAM is that the KL regularisation is too restrictive, which results in less informative representations compared to LIAM. To alleviate this issue, we can use a $\beta$ coefficient to weigh the KL loss, similarly to Higgins et al. [2017] or use the $MMD^2$ distance [Gretton et al., 2007] for regularisation, similarly to info-VAE [Zhao et al., 2017]. In both cases the resulting objective is not a lower bound to the evidence. We did not find any important benefit of using a $\beta$ coefficient or the $MMD^2$ distance compared to the simpler instantiation of LIAM that we present in this work.

## 4.6    Understanding LIAM

To better understand the working mechanisms of LIAM, we analyse how LIAM performs agent modelling in one episode of the double speaker-listener environment (shown in Figures 6a to 6c). In this episode, the controlled agent has colour red, while the modelled agent has colour blue. The reconstruction of the colour of the controlled agent can be seen as a proxy of the belief about its colour, that the controlled agent encodes into $z$ at each time step. Note that the reconstructed values presented in Figure 6d are not a probability distribution (can be lower than 0 or larger than 1) due to unconstrained optimisation, but we consider that the value closest to 1 represents the belief of the controlled agent about its colour. At the time step 0, all reconstructed colours have similar value close to 0.3 meaning that the controlled agent does not have any information about its colour. The controlled agent outputs different messages and observes how the modelled agent reacts to them through the episode. We observe that at time step 7 the controlled agent believes that its colour is blue and starts moving toward the blue landmark. After the 10th interaction, the controlled agent has communicated several different messages to the modelled agent and by observing how the modelled agent reacts to them, it understands that its belief about its colour was wrong, and it starts moving toward the correct landmark.

## 5    Conclusion

We proposed LIAM, the idea of learning models about the trajectories of the modelled agent using the trajectories of the controlled agent. LIAM concurrently trains the model with a decision policy, such that the resulting agent model is conditioned on the local observations of the controlled agent. LIAM is agnostic to the type of interactions in the environment (cooperative, competitive, mixed) and can

model an arbitrary number agents. Our results show that LIAM can significantly improve the episodic returns that the controlled agent achieves over baseline methods that do not use agent modelling. The returns that are achieved by A2C when combined with LIAM in some cases nearly match the returns that are achieved by the upper baseline FIAM which assumes access to the trajectories of the modelled agent during execution.

We identify two main limitations of LIAM that are left unaddressed: (a) modelling non-reactive agents, and (b) modelling agents from high-dimensional observations such images. First, a necessary requirement for successfully applying LIAM is for the actions of the modelled agent to affect the observations of the controlled agent. If this assumption does not hold, LIAM will not have enough information to reason about the modelled agent's trajectory and will just learn an average representation over the possible trajectories of the modelled agent. Second, in this work, we focus on environments with low-dimensional vector-based observations. Recent works on state representation for RL in images [Laskin et al., 2020, Zhang et al., 2020] have shown that reconstruction-based methods may learn low-quality representations because they weigh the reconstruction of each pixel as equally important even though the vast majority of the pixels contain no relevant information to the modelling problem. Extending LIAM to pixel-based observations can be achieved by using a different architecture for extracting representations, such as CARL.

In the future, we would like to investigate how agent modelling from local information can be extended to multi-agent reinforcement learning, where several agents are learning concurrently and non-stationarity arises [Hernandez-Leal et al., 2017, Papoudakis et al., 2019]. We would also like to explore notions of "safety" to handle agents that aim to deceive and exploit the agent model [Shoham et al., 2007, Ganzfried and Sandholm, 2011, 2015].

## Funding Disclosure

This research was in part supported by the UK EPSRC Centre for Doctoral Training in Robotics and Autonomous Systems (G.P., F.C.), and personal grants from the Royal Society and the Alan Turing Institute (S.A.).

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
