# A Pseudocode of LIAM

Algorithm 1 shows the pseudocode of LIAM.

---
**Algorithm 1** Pseudocode of LIAM's algorithm
---

   **for** $m = 1, ..., M$ episodes **do**
      Reset the hidden state of the encoder LSTM
      Sample $E$ fixed policies from $\Pi$
      Create $E$ parallel environments and
      gather initial observations
      $a_{-1}^1 \leftarrow$ zero vectors
      **for** $t = 0, ..., H - 1$ **do**
         **for** every environment $e$ in $E$ **do**
            Get observations $o_t^1$ and $o_t^{-1}$
            Compute the embedding $z_t = f_w(o_t^1, a_{t-1}^1)$
            Sample action $a_t^1 \sim \pi(a_t^1 | o_t^1, z_t)$
            Sample modelled agent's action $a_t^{-1}$
            Perform the actions and get $o_{t+1}^1, o_{t+1}^{-1}, r_{t+1}^1$
         **end for**
         **if** $t \mod$ update_frequency $= 0$ **then**
            Gather the sequences of all $E$ environments in a single batch $B$
            Perform a gradient step to minimise $\mathcal{L}_{ED}$ (Equation (2)) using $B$
            Perform a gradient step to minimise $\mathcal{L}_{A2C}$ (Equation (3)) using $B$
         **end if**
      **end for**
   **end for**

---

# B Fixed Policies

**Double Speaker-Listener:** Each policy consists of two sub-policies: the communication message policy and the navigation action policy. For the communication message policy, we manually created constant five-dimensional one-hot communication messages that correspond to different colours. We manually select different fixed policies that communicate the same colours with different communication messages. For the navigation policies, we then created five pairs of agents and we trained each pair to learn to navigate using the MADDPG algorithm [Lowe et al., 2017]. Each agent on the pair learns to navigate based on the communication messages of the other agent in the pair.

**Level-Based Foraging:** The fixed policies in level-based foraging consist of four heuristic policies, three policies trained with IA2C and three policies trained with MADDPG. All modelled agent policies condition their policies to the state of the environment. The heuristic agents were selected to be as diverse as possible, while still being valid strategies. We used the strategies from Albrecht and Stone [2017], which are: (i) going to the closest food, (ii) going to the food which is closest to the centre of visible players, (iii) going to the closest compatible food, and (iv) going to the food that is closest to all visible players such that the sum of their and the agent's level is sufficient to load it. We also trained three policies with MADDPG by training two pairs of agents and extracting the trained parameters of those agents. For the policies trained with MADDPG, we circumvent the instability caused by deterministic policies in Level-based Foraging by enabling dropout in the policy layers [Gal and Ghahramani, 2016] both during exploration and evaluation. We observe that by creating stochastic policies the agents perform significantly better.

**Predator-Prey:** The fixed policies in the predator-prey consist of a combination of heuristic and pretrained policies. First we created four heuristic policies, which are: (i) going after the prey, (ii) going after one of the predators, (iii) going after the agent (predator or prey) that is closest, (iv) going after the predator that is closest. Additionally, we create another six pretrained policies, by training two sets of three agents using RL: one with the MADDPG algorithm and one with the IA2C algorithm. To create the ten fixed policies, where each fixed policy consists of three sub-policies (one for each predator), we randomly combine the four heuristic and the six pretrained policies.

## C  Baselines

### C.1  Contrastive Agent Representation Learning (CARL)

CARL extracts representations about the modelled agent in the environment without reconstruction from the local information provided to the controlled agents, the locally available observation, and the action that the controlled agent previously performed. CARL has access to the trajectories of all the other agents in the environment during training, but during execution only to the local trajectory.

To extract such representations, we use self-supervised learning based on recent advances on contrastive learning [Oord et al., 2018, He et al., 2020, Chen et al., 2020a,b]. We assume a batch $B$ of $M$ number of global episodic trajectories $B = \{\tau^{glo,m}\}_{m=0}^{M-1}$, where each global trajectory consists of the trajectory of the controlled agent 1 and the trajectory of all the modelled agent $-1$, $\tau^{glo,m} = \{\tau^{1,m}, \tau^{-1,m}\}$. The positive pairs are defined between the trajectory of the controlled agent and the trajectory of the modelled agent at each episode $m$ in the batch. The negative pairs are defined between the trajectory of the controlled agent at the specific episode $m$ and the trajectory of the modelled, in all other episodes $l \neq m$ in the batch, expect episode $m$.

$$
\begin{aligned}
\text{pos} &= \{\tau^{1,m}, \tau^{-1,m}\} \\
\text{neg} &= \{\tau^{1,m}, \tau^{-1,l}\}
\end{aligned}
\tag{4}
$$

We assume the existence of two encoders: the recurrent encoder that receives sequentially the trajectory of the controlled agent $f_{\boldsymbol{w}} : \tau^1 \to \mathcal{Z}$ and at each time step $t$ generates the representation $z_t^1$, and the recurrent encoder that receives sequentially the trajectory of the modelled agent $f_{\boldsymbol{u}} : \tau^{-1} \to \mathcal{Z}$ and at each time step $t$ generates the representation $z_t^{-1}$. The representation $z_t^1$ is used as input in the actor and the critic of A2C. During training and given a batch of episode trajectories we construct the positive and negative pairs following Equation (4) and minimise the InfoNCE loss [Oord et al., 2018] that attracts the positive pairs and repels the negative pairs.

$$
\mathcal{L}_{CARL} = -\sum_{t=0}^{H-1} \log \frac{\exp\{\cos(z_t^{1,m}, z_t^{-1,m})/\tau_{temp}\}}{\sum_{j=0}^{M-1} \exp\{\cos(z_t^{1,m}, z_t^{-1,j})/\tau_{temp}\}}
\tag{5}
$$

where cos is the cosine similarity and $\tau_{temp}$ the temperature of the softmax function.

### C.2  LIAM-VAE

Following the work of Chung et al. [2015] we can write the lower bound in the log-evidence of the modelled agent's trajectory as:

$$
\log p(\tau^{-1}) \geq \mathbb{E}_{z \sim q(z|\tau^1)} \left[ \sum_t [\log p(\tau_t^{-1}|z_t, \tau_{:t-1}^{-1}) - D_{\mathrm{KL}}(q(z_t|z_{:t-1}, \tau_{:t}^1) \| p(z_t|\tau_{:t-1}^1, z_{:t-1})) \right]
\tag{6}
$$

We assume the following independence $\tau_t|z_t \perp\!\!\!\perp \tau_{:t-1}$. This practically means that the latent variables should hold enough information to reconstruct the trajectory at time step $t$. We deliberately make this assumption that will lead to worse reconstruction but more informative latent variables. Since the goal of LIAM-VAE is to learn representation about the modelled agent we prioritise informative representations over good reconstruction. More specifically, consider that we want to reconstruct modelled agent's observation $o_t^{-1}$ at time step $t$, using the latent variable $z_t$. The observation of the modelled agent $o_{t-1}^{-1}$ has as information the colour of the controlled agent. If we condition the decoder both on the latent variable and the previous observation to reconstruct the current observation, then the reconstruction of the colour of the controlled agent can be achieved by the decoder by looking at the observation $o_{t-1}^{-1}$. As a result, the latent variables will not encode this type of information that is necessary to successfully solve this environment. Additionally, we assume that $\tau_t|\tau_{:t-1} \perp\!\!\!\perp z_{t-1}$. This assumption holds because $z_{t-1}$ is generated from a distribution conditioned on $\tau_{:t-1}$ and $z_{t-1}$ holds the same information as $\tau_{:t-1}$. As a result, we can write the lower bound as:

$$
\log p(\tau^{-1}) \geq \mathbb{E}_{z \sim q(z|\tau^1)} \left[ \sum_t [\log p(\tau_t^{-1}|z_t) - D_{\mathrm{KL}}(q(z_t|\tau_{:t}^1) \| p(z_t|\tau_{:t-1})) \right]
\tag{7}
$$

To optimise this lower bound, we define:

- The encoder $q_{\boldsymbol{w}}$ with parameters $\boldsymbol{w}$, which a recurrent network that receives as input the observations and actions of the controlled agent sequential and generates the statistics (the mean and the logarithmic variance) of a Gaussian distribution.

- The decoder $p_{\boldsymbol{u}}$ with parameters $\boldsymbol{u}$ that receives the latent variable $z_t$ and reconstructs the modelled agent's trajectory at each time step $t$. The implementation of the decoder is the same as the decoder of LIAM described in Equation (2).

- The prior $p_{\boldsymbol{\phi}}$ with parameters $\phi$ that models the temporal relationship between the latent variables. We evaluated two different models for the prior: one that receives the hidden state of the recurrent network of the encoder as input and outputs a Gaussian distribution, and one that considers that the prior at each time step $t$ is equal to the posterior at each time step $t - 1$. Both prior choices led to similar episodic returns. In Figure 5 we present the average returns for the later choice of prior.

We train LIAM-VAE similarly to LIAM. In the actor and the critic network we input the mean of the Gaussian posterior.

## D   Implementation Details

All feed-forward neural networks have two hidden layers with ReLU [Maas et al., 2013] activation function. The encoder consists of one LSTM [Schmidhuber and Hochreiter, 1997] and a linear layer with ReLU activation function. All hidden layers consist of 128 nodes. The action reconstruction output of the decoder is passed through a Softmax activation function to approximate the categorical modelled agent's policy. For a continuous action space, a Gaussian distribution can be used. For the advantage computation, we use the Generalised Advantage Estimator [Schulman et al., 2015] with $\lambda_{GAE} = 0.95$. We create 10 parallel environments to break the correlation between consecutive samples. The actor and the critic share all hidden layers in the A2C implementation. We use the Adam optimiser [Kingma and Ba, 2015] with learning rates $3 \times 10^{-4}$ and $7 \times 10^{-4}$ for the A2C and the encoder-decoder loss respectively. We also clip the gradient norm to $0.5$. We subtract the policy entropy from the actor loss [Mnih et al., 2016] to ensure sufficient exploration. The entropy weight $\beta$ is $10^{-2}$ in double speaker-listener and the predator-prey and $0.001$ in the level-based foraging. We train for 40 million time steps in the double speaker-listener environment and for 10 million time steps in the rest of the environments. During the hyperparameter selection, we searched: (1) learning rates in the range $[10^{-4}, 7 \times 10^{-4}]$ and $[3 \times 10^{-4}, 10^{-3}]$ for the parameters of RL and LIAM respectively, (2) hidden size between 64 and 128, and (3) entropy regularisation in the range of $[10^{-3}, 10^{-2}]$. The hyperparameters were optimised in the double speaker-listener environment and were kept constant for the rest of the environments, except the entropy regularisation in level-based foraging, where we saw significant gains in the performance of all algorithms.

For the MPE environments, we use the version that can be found in this link: `https://github.com/shariqiqbal2810/multiagent-particle-envs`. This version allows for better visualisation of the communication messages in the double speaker-listener environment, as well as seeding the environments. Our A2C implementation is based on the following well-known repository: `https://github.com/ikostrikov/pytorch-a2c-ppo-acktr-gail/tree/master/a2c_ppo_acktr`.

## E   Different Learning Rates

In this Section, we evaluate the final achieved returns of LIAM with respect to different learning rates for the RL and the encoder-decoder optimisation. Figure 7 presents a heat-map of LIAM's achieved returns in the double speaker-listener environment with respect to different combinations of the two learning rates. We observe that LIAM's achieved returns are close to the returns achieved by the best configuration for most of the evaluated learning rate configurations.

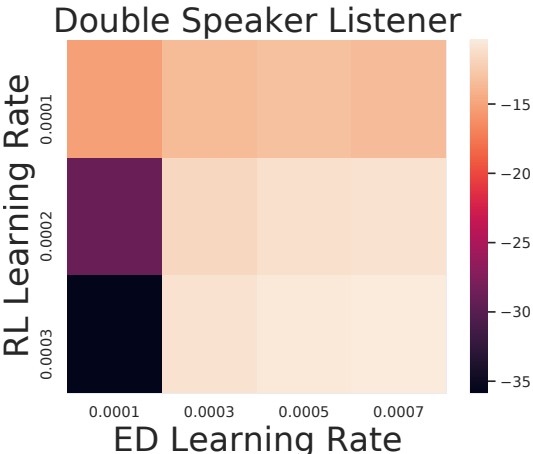

Figure 7: Heat-map with different learning rates for the RL and the encoder-decoder optimisation.

## F    Scalability in the Number of Fixed Policies

In this section we evaluate LIAM with respect to 100 different fixed policies. Instead of combining the two sub-policies in the double speaker-listener environment in one-to-one fashion, we take the Cartesian product of the sub-policies which leads to 100 different combinations of fixed policies.

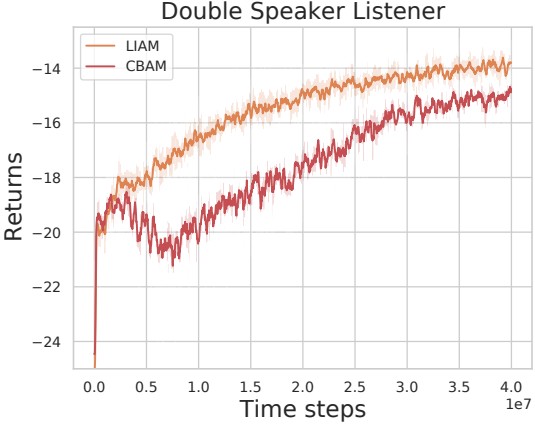

Figure 8: Episodic evaluation returns and $95\%$ confidence interval of LIAM and CBAM during training, against 100 fixed policies.

Figure 8 presents the average returns achieved by LIAM and CBAM against 100 different fixed policies. In Section 4.3, we observed that in the double speaker-listener environment the average returns of CBAM were very close to the returns of LIAM. We observe that with 100 fixed policies the difference in the returns between LIAM and CBAM increases significantly. This does not come as a surprise, since CBAM would require a very flexible policy to successfully adapt to all 100 fixed policies. On the other hand, as we observed in Figure 4a the embeddings of LIAM are not explicitly clustered based on the identities of the fixed policies, but based on trajectory of the modelled agent to encode its communication message and its observation. As long as the embeddings contain such information, the RL procedure is able to learn a policy that achieves higher returns compared to CBAM.