# OpenReview forum: "Agent Modelling under Partial Observability for Deep Reinforcement Learning"
_NeurIPS.cc/2021/Conference — NeurIPS 2021 Poster_

### Official Review · Reviewer_qnRK · 2021-07-05

**Rating:** 7
**Confidence:** 3

**Summary:**

This paper proposes an encoder-decoder type neural network for agent modelling in partially observable domains. The method seeks to learn a useful encoding of a controlled agent's action-observation histories to a latent vector. By useful, it is meant that the learned representation used as an input to a downstream RL method allows finding "best response" policies to a set of fixed opponent/other agent policies. At training time, decoders are trained to predict the observations and actions of the other agents. At test time, the decoder is discarded, and the encoder part is used to output latent vectors. The proposed approach is demonstrated in three MARL domains, showing improvement over baselines such as simply classifying other agents' policies, or discarding the agent information completely. Ablation studies confirm the effectiveness of the design choices for the proposed method.

**Limitations And Societal Impact:**

Limitations of the work are appropriately addressed. Paper does not discuss societal impact.

**Main Review:**

# Positives

- As measured by the returns, the proposed method has good empirical performance in the domains considered. It improves on several baselines, that seem to be well chosen. It is also interesting to see that simply identifying which of the fixed policies the other agents are using (CBAM) has very limited usefulness, even in simple domains.

- The ablation studies are very nice, showing that the components of the proposed method are indeed useful. The encodings learned are reasonable in the speaker-listener domain. The ablations also clearly make the point that the entire trajectory of actions and observations matter in a partially observable setting.

- It is nice to see that the paper tackles the challenging partially observable setting, rarely considered in other works in agent modelling I am aware of. The paper is very well written and organized, and provides a good overview of recent agent modelling methods.

# Limitations

1. The set of possible opponent policies is assumed fixed. This limits the usefulness of the proposed approach somewhat, at least in competitive settings it is reasonable to expect some adaptation from the opponents.

2. As acknowledged by the authors (Appendix D), the method as presented is unlikely to scale beyond low-dimensional observation/action spaces, as it requires predicting the complete actions and observations of the other agents. It is not obvious to me how representation learning methods could help here.

3. In the domain studied in the paper, the proposed method works well. What could be an interesting future addition is to find some failure cases, which could point out the limits or perhaps features of the domain that make it challenging to apply this type of agent modelling.

4. The proposed LIAM method can be seen as a type of extra regularization, which could also partly explain the authors' observation of its stability. The assumption of this paper is that it is not enough to learn any representation of the controlled agents' action-observation history, but it should be one that enables prediction of the other agents' actions/observations. This suggests a potential additional weakness for the proposed method; in the presences of nuisance/unnecessary information, the method might overregularize to be able to predict useless details of the other agents' actions/observations. This complements point 2. above, and can appear even in low-dimensional settings.


Other comments:

- I was initially a bit confused why the POSG is chosen as a formalism, since the paper assumes that the policies of the other agents are fixed. In fact, the problem being solved is a POMDP, specifically searching for the best response policy for the controlled agents while other agents act according to some fixed policies randomly drawn from a set of policies.

- In the ablation study, the VAE method does not seem to help much. The reason stated is the restrictiveness of the KL regularization. But given that the messages received can uniquely be mapped to a specific fixed policy, it seems that there is generally not much to gain from a stochastic VAE? Did you observe the VAE to be helpful in the other domains?

- Line 117 defines an independent observation model for each agent. But I did not see anything that prevents applying this method when the observation model specifies a joint distribution over all agents' observations, so this seems like an opportunity for a small additional generalization.

- Line 87: remove extra "sample"

# Post rebuttal
I thank the authors for their response, in particular for clarifying their ideas regarding representation learning. Overall, this is a nice paper with some interesting observations and contributions.

**Time Spent Reviewing:**

3

---

> ### Author Response · Authors · 2021-08-10
> **Response to reviewer qnRK**
>
>
> We would like to thank reviewer qnRK for their review.
>
> ### Limitations
>
> 1. We point the reviewer to our response to reviewer 5w5T (point 2)
>
> 2. Representation learning for pixel-based observations has been researched for many years by the community, and several recent works are still focusing on reconstruction-based representation learning [1, 2]. Additionally, recent works [3, 4] have been proposed that use non-reconstruction based representation learning, such as contrastive learning to avoid issues that come from reconstructing high-dimensional observations as pointed out by the reviewer. Therefore, we believe that the selection of the appropriate representation learning method is of utmost importance, and that representation learning is an exciting avenue of future research.
>
> 3.  We present an example environment in Appendix D. Here, LIAM is unable to learn any informative representations about the fixed policies. The reason is that the modelled agent is not reactive (and it does not alter the observation of the controlled agent during the episode), and as a result, LIAM is unable to infer hidden information that only exists in the modelled agent's observations.
>
> 4. We agree with the reviewer's comment regarding the prediction of irrelevant information about the modelled agent, and this is an open problem in representation learning. Domain knowledge can be used to limit this issue, especially in low dimensional environments. For example, in DSL, we could force LIAM to only predict the hidden colour or only some parts of the modelled agent's observations, and that would make the task easier.
>
>
> ### Other comments:
>
> * We agree with the reviewer that a POMDP can be used instead of a POSG. We use the POSG framework to define the presence of other agents in the environment.
>
> * We did not find the VAE to be much of a help. We speculate that in pixel-based observation, a VAE or a beta-VAE could be used to create disentangled representations and that could lead to higher achieved returns by the RL policy.
>
> * We agree with the reviewer's comment. A joint distribution over the observations of all agents can be defined. From the point of training  LIAM, all modelled agents in the environment are perceived as a single agent, with a joint observation and action.
>
> * We would like to thank the reviewer for pointing us to this typo. We will fix that in the future revision of the paper.
>
> [1] David Ha et. al. Recurrent World Models Facilitate Policy Evolution. NeurIPS 2018.
>
> [2] Hafner et al. Mastering Atari with Discrete World Models. ICLR 2021.
>
> [3] Laskin et al. Contrastive Unsupervised Representations for Reinforcement Learning. ICML 2020
>
> [4] Zhang et al. Learning Invariant Representations for Reinforcement Learning without Reconstruction. ICLR 2021

---

### Official Review · Reviewer_en5v · 2021-07-09

**Rating:** 6
**Confidence:** 4

**Summary:**

The authors introduce a method for learning in a multi-agent setting that centers around predicting what other agents will do in the given context. Crucially, global information is not assumed (at execution), and so the techniques introduced have a focus on agent perspective in the information used to drive behaviour.

The methods are evaluated on three domains requiring multi-agent consideration (coordination or combativeness), and the results demonstrate the merit in taking the local perspective for these partial-information domains.

**Ethical Concerns:**

I see no substantial ethical concerns with the paper/research.

**Limitations And Societal Impact:**

There are some substantial limitations/assumptions on the approach, and the authors address some of them. At this stage in the research, I don't believe there are any major societal impact concerns to address.

**Main Review:**

## Originality

The problem setting is certainly interesting. Assumptions of full observability and perspective of all agents, by all agents, is certainly far too strong of an assumption. This work takes a reasonable step to relax that.

The one area that seems to be lacking from the related work is the growing body on embodied agents. There are a mix of papers at various conferences, but these two workshop series seem directly aligned:
- https://embodied-ai.org/
- https://eyewear-computing.org/EPIC_CVPR21/past-editions


## Quality
The paper is well written, and the quality of the work is good. My biggest complaint on the framing of the solution (which also is an issue with the significance) is the idea that _the perspective of the other agents is available, in full, during training_. Projecting (ala Theory of Mind) during the training process (and evaluation, for that matter) seems to be a far more realistic avenue than the strong assumption that you have access to full agent trajectories when training.

## Clarity
The paper is generally very clear. There are a few things that can be improved, however:

1. Lines 45-47 are confusing, and only made sense after having read the full paper at least once.
2. The plots are often too small to be legible (e.g., Fig 4).
3. You should emphasize from the beginning that this is a multi-agent setting, and from the perspective of one agent. Partial observability exists in the single-agent setting, and your work is more ambitious than just that.
4. The "modelled agent" used throughout is actually capturing several (all) other agents. I would suggest changing the language here, as it feels as though you are only considering 2-player games.


## Significance
There are a fair number of assumptions that come into play with this work, and that affects the significance substantially. I applaud the authors for trying to move to the ego-centric setting, as it is far more realistic than the omnipotent view many multi-agent approaches assume, but the assumption of full observability during training remains a very strong one. It is clear how much this assumption helps, given the poor performance of LIAM-Local (a more realistic setup for what a controlling agent would have access to).

Another one of the assumptions is that the other agents' behaviour depends entirely on the working memory of the controlled agent. There should be elements of what the other agents do/see/experience that influence their behaviour, and in no way is something the controllable agent sees. This is touched upon by the authors as the limitation of needing to be roughly local (so some level of mutual observability is maintained), but manifests as a questionable architecture choice (having the predicted $o_t^{-1}$ based entirely on $z_t$). I get that the other agents' knowledge may be captured somehow in $z_t$, but the only vector for this is through $o_t^{1}$, which feels like a major limitation to me.

The final major limitation is with the complexity of behaviour on the other agents. There is little justification as to how realistic having pre-determined policies might be, and even with just a handful of individual policies or actions availability, the cross-product in action-space of the collective "other agent(s)" can quickly grow out of hand. It's unclear why you need the assumption of a fixed set of policies, rather than simply attempting to learn the function mapping the (projected) observation space of the other agents to their actions.

## Post-Rebuttal Comments
It seems as thought my final concern on policy representation is in fact not an issue. I would recommend that the authors make it clear in the intro and section 3.1 text that the fixed policies are not something the local agent has access to.

**Time Spent Reviewing:**

5

---

> ### Author Response · Authors · 2021-08-10
> **Response to reviewer en5v**
>
> We would like to thank reviewer en5v for their review.
>
> ### Originality
>
> We would like to thank the reviewer for pointing us to these workshop series. We will include more references in the next version of the paper.
>
> ### Clarity
>
> We would like to thank the reviewer for pointing us to these clarity issues. We will address them in the next version of the paper.
>
> ### Significance
>
> We agree that the long-term goal of agent modelling research should be to learn models based only on the local observations and actions of the controlled agent, and it would be an important limitation if the training was done in real-world settings and not a simulator.
> The use of the modelled agent's trajectory is in line with other research areas in (simulation-based) multi-agent systems, such as the centralised training decentralised execution paradigm in multi-agent RL, as discussed in Section 2. Since training is performed in a simulator, sharing the observations between agents does not cause significant computational overhead.
>
> We agree with the reviewer that the reconstruction task (trying to predict $o^{-1}_t$ given $z_t$) in some cases is infeasible, because the modelled agent is out of the receptive field of the controlled agent, and in this case, the z will be uninformative and the decoder will make an "average" prediction over the possible observations and actions.
> However, in most environments the sequence of local observations provides the controlled agent with "hints", and using the recurrent encoder of LIAM, it can learn to (even partially) reconstruct the modelled agent's trajectory.
> We also point the reviewer to Section 4.4, where we present how the accuracy of hidden information reconstruction changes with respect to the number of time steps in the episode.
>
> Regarding the complexity of the fixed set of policies: In DSL, the fixed policies are very diverse and significantly different between them. In LBF and PP, we combine several heuristic and pretrained policies. In these two environments, the fixed policies can be similar at some time steps in the episode. Even in this case, LIAM can make informative decisions that lead to better returns compared to the other baselines, by inferring information about their hidden observations.
>
> Regarding the fixed policies: We point the reviewer to our response to reviewer 5w5T (point 2)

---

> > ### Comment · Reviewer_en5v · 2021-08-13
> > **Rebuttal Response**
> >
> > ### Observability during training
> >
> > I'm less concerned about the computational overhead as I am with the shift in distribution for the agent. Training is done with an entirely different set of assumptions. You could, for example, force an egocentric view of the learning agent in training and still appeal to a simulator. Having access to a simulated setting, and reasonable approximations of other agent's behaviour, are still assumptions at play in this alternative training regime. But at least you don't expose unrealistic signals to the learning agent.
> >
> > ### Information leakage through $z_t$
> >
> > Well it seems to cut in two ways: (1) there is information that the other agents would observe and is absent from $z_t$ (point taken that the learned agent hopefully averages the trajectories here); and (2) information can pass from the egocentric agent's observations, via $z_t$ to what the other agent's "see". This is all in the "eyes" of the acting agent, and so I realize at execution time that $z_t$ isn't communicated directly to the other agents, but it really conflates what the acting agent thinks the other agents are capable of seeing.
> >
> > What I think is missing here is some notion of the perspective shift from the learned agent to the other agents in the environment, and I'm not convinced that the current architecture is equipped to learn this epistemic detail.
> >
> > ### Policy assumptions
> >
> > My concern here should be clarified -- it wasn't a matter of having a (non-)stationary property, but rather having them enumerated in advance. How would the approach be modified to handle fixed-but-unknown policies of the other agents?

---

> > > ### Author Response · Authors · 2021-08-18
> > > **Response to reviewer en5v**
> > >
> > > Thank you for your response.
> > >
> > > ### Observability during training
> > >
> > > Generalisation under a large distribution shift is an open problem in machine learning, and not much progress has been achieved in RL. We agree with the reviewer that there can be a large shift in the distribution if the fixed policies during training are significantly different from the fixed policies during evaluation. However, this depends entirely on the properties of the environment and the fixed set of policies. For example, as we mention in our response to reviewer 5w5T (point 2), in DSL, we cannot expect the learning agent to generalise to unseen fixed policies, due to the large shift in the distribution. In other environments, generalising to unseen fixed policies is possible as we show in our response to reviewer 5w5T (point 2). Additionally, a large distribution shift can be created even in the case where we train LIAM solely with the trajectories of the controlled agent. We do not aim to claim that distribution shift is not an issue with LIAM, but this shift is an issue in most ML systems. Finally, the main goal of this paper, as we mention in lines 37-39 is to learn the relationship between the trajectory of the controlled and the modelled agent.
> > >
> > > ###  Information leakage through $z_t$
> > >
> > > The perspective shift is achieved through the encoder-decoder module. The encoder receives sequentially the trajectory of the controlled agents and learns how this corresponds to the trajectory of the modelled agent. This is the main idea behind LIAM; an algorithm that can infer the hidden trajectory of the modelled agent using the trajectory of the controlled agent.
> > >
> > > ### Policy assumptions
> > >
> > > Our experimental framework requires the existence of a set of policies for the modelled agent. However, from the perspective of LIAM, the fixed policies are not enumerated. LIAM does not know the number of fixed policies or the identity of the fixed policy that is used by the modelled agent (neither during training nor execution). We also show that generalising to unseen fixed policies is possible (we point the reviewer to our response to reviewer 5w5T (point 2)) as long as the distribution shift between the training fixed policies and the unknown fixed policies is not large.

---

> > > > ### Comment · Reviewer_en5v · 2021-08-19
> > > > **Not such a big assumption...**
> > > >
> > > > I may have again mislead with the far-too-loose phrase "_I'm less concerned about the computational overhead as I am with the shift in distribution for the agent._". I was referring to the shift in what the agent actually has access to -- observations for the other agents -vs- just the local one. The difference between the train and eval setup remains a leap conceptually for me.
> > > >
> > > > ### Re: Policy Assumptions
> > > > I was misled by how 3.1 was presented and this early statement ("given" here isn't referring to access of the local agent):
> > > > > We assume a given set of possible policies for the non-learning agents and that these policies are fixed.
> > > >
> > > > This is indeed a far more appropriate assumption. I will update by assessment to reflect this, and make the recommendation that these details be emphasized.

---

### Official Review · Reviewer_Nuzh · 2021-07-16

**Rating:** 6
**Confidence:** 4

**Summary:**

This paper introduces the Local Information Agent Modelling (LIAM), which learns a latent representation of the other agents. By using an encoder-decoder structure, the latent inference could be performed only based on the ego agent's local information, without knowing the observations and actions of other agents during the test time. The learned latent state is then used to augment the observation of the ego agent for reinforcement learning.  The experiment results show the advantage of LIAM compared with other baselines tested.

**Ethical Concerns:**

I don't find any ethical issues.

**Limitations And Societal Impact:**

The limitations discussed by the authors in the appendix are important, and the potential solutions mentioned are reasonable.

**Main Review:**

The contributions of the paper, to my knowledge, are novel. The method proposed is intuitive and well-explained by the authors. The experiment results and discussions deliver convincing evidence on the advantage of the proposed method. The paper is well-written. The problem that the paper aims to address is important since, in many practical multi-agent problems, the actions and observations of other agents are unknown to the ego agent.

There are some minor concerns about the paper:
First, simultaneously training the policy and the encoder where the two modules interact with each other may cause training stability issues even the policy is trained under an off-policy RL algorithm. In the paper, the author mentions that they are using different learning rates for the two modules. It would be good if there are some more studies and explanations on this issue.
Second, in the ablation study, the author compares the different encoders. I think maybe some move ablation studies on the decoder are also important. For example, instead of predicting both the observation and the action, how would it influence the performance of the encoder only predicts one of them?
Third, the assumption of a fixed finite set of other agents' policies limits the application of the method and seems unnecessary. Some more extensions would be good.
Fourth, it would be great if the authors could also discuss the scalability of the method.

**Time Spent Reviewing:**

3

---

> ### Author Response · Authors · 2021-08-10
> **Response to reviewer Nuzh**
>
> We would like to thank reviewer Nuzh for their review.
>
> Re "First": we did not find LIAM to be unstable in general. Empirically we found that encoder-decoder methods tend to be more stable during training compared to RL methods. Therefore, to speed up learning, we chose to use a larger learning rate for training the agent model compared to the policy. We will add experiments where we experiment with different learning rates in the appendix of the paper.
>
> Re "Second": from some initial experiments, we have found that reconstructing the observation of the modelled agent is more important than reconstructing its actions in the environments that we used for evaluation. We will add these two ablated baselines (one for reconstructing only the observations, and one for only reconstructing the actions) in Section 4.5.
>
> Re "Third": please see our response to reviewer 5w5T (point 2).
>
> Re "Fourth": we test the scalability of LIAM with respect to the number of modelled agent policies in Appendix F.
> The dimensions of the outputs of LIAM's decoders increase linearly with respect to the number of agents since the dimensions of the joint observation and action also increase linearly with respect to the number of agents (we also point the reviewer to our response to reviewer 5w5T (point 3)).

---

### Official Review · Reviewer_5w5T · 2021-07-16

**Rating:** 6
**Confidence:** 3

**Summary:**

The authors propose "local information agent modeling" (LIAM), i.e. using a supervised auxiliary predictive task to help learn a representation of the history which is useful to predict other agent's behaviors, and which is then used as a basis for decision making.  The task encoded by LIAM involves predicting the modeled agents' trajectories based on the controlled agent's trajectory.  LIAM is presented both as a general idea, and as a specific implementation which uses encoder-decoder models (controlled agent trajectory -> latent representation -> modeled agent trajectory).  The auxiliary learning task can be performed with supervision in an offline learning framework, where the privileged information of all agents is available.  During execution, the fully trained encoder can be used without requiring the privileged information.

The experiment section involves 3 multi-agent environments, and 5 baselines closely related to LIAM.  On top of the standard performance-based RL results, the authors include a) a model evaluation section where they analyze the learned latent representations and their ability to encode important information, and b) an ablation study which shows the empirical importance of multiple individual components of their specific LIAM implementation.

**Ethical Concerns:**

As above, the work is fairly generic and does not intrinsically raise any specific concerns which wouldn't be shared by the RL field as a whole.

**Limitations And Societal Impact:**

The submission addresses fairly generic multi-agent RL settings, rather than specific applications.  I do not believe this work warrants any specific statements about societal impact.

**Main Review:**

EDIT:  The author's response addresses my primary concerns regarding fairness, so I've updated my score based on their response.  (more updates may come based on the committee discussion).

---

The work seems like a solid advancement in the field of multi-agent control, and the authors have positioned it well in the current literature.  The (offline self-)supervised task encoded by LIAM is fairly simple and perhaps in retrospect an obvious one (which I consider a strength!).  While it is a variant of other similar auxiliary learning tasks, LIAM does address significant drawbacks made by prior work (e.g. full online observability, or (partial) knowledge of the domain model).  In contrast, LIAM primarily makes use of a single but fairly common assupmtion (two more are stated, but it is not clear to me that they are necessary.  More on this later):  the ability to have offline training in a controlled environment where privileged information is available.  The task encoded by LIAM makes ample sense in the specific context of multi-agent RL problems which *require* some form of interaction between agents.  As a bonus, LIAM makes sense in all sorts of generic stochastic multi-agent games, i.e., cooperative, competitive, or mixed games.  The model evaluation and ablation study sections are very welcome additions which complement well the standard performance-based RL results.

Nonetheless, both LIAM and the evaluation performed by the authors come with some limitations and lingering questions which in my mind need to be addressed.  My current rating is based on the current submission as-is, under my current understanding and the temporary assumption that there will not be an author response;  I will gladly adjust my rating based on the author's response.

In order of importance:

1) Some baseline methods use RNNs which receive observation-action sequences (CBAM), while others also receive the rewards as input (NAM and VariBad).  For some methods, it is unclear whether rewards are used or not (LIAM,FIAM,CARL).  I hope this is simply a mistake in the descriptions of the methods, but will pose my question as if the descriptions were accurate.  This setup is very problematic for multiple reasons:

1.a) Such an inconsistency between the information available to each method/baseline wholly undermines the empirical evaluation.  All methods/baselines should access the same information (in principle, either all should observe the rewards, or all should not observe the rewards).  In the general case, I believe that observable rewards are a problem on their own right, as described next.

1.b) "Observable" rewards (which are fed into the RNN and influence the choice of future actions) are IMO extremely problematic in partially observable control problems.  The rewards contain information about the underlying state which are not available from the observations alone.  This not only significantly alters our understanding of the problem, but in some cases may allow the agents to significantly bypass the need to develop information-gathering and communication strategies.  E.g., in the DSL environment, being able to observe the reward encodes the agent distances from their goal, and provides a strong signal to be able to locate the landmarks without the need to communicate.

2) The assumption that other agent policies are taken from a fixed set of stationary policies which does not change over time seems to be a relatively strong one;  however, it is not clear to me that this assupmtion is intrinsically necessary.  Is there any reason why this assumption is made, aside from making the RL and LIAM tasks easier?  A natural idea would be to run evaluations where all agents are concurrently trained using LIAM;  any reason why this evaluation was not performed?

3) Treating the policies and behavior of all other agents as a monolithic joint entity is perhaps the biggest limitation of the current implementation as a whole, and one which I must assume makes LIAM scale poorly to the number of agents.  Given that the evaluation environments involve only 1 or 3 other modeled agents, it certainly is not sufficient to confirm generous statements like "LIAM [...] can model an arbitrary number of agents".  Is it possible to factorize the predictions of modeled agent trajectories?

4) Another limitation is the assumption that the modeled agent policies are reactive, i.e. that the next actions are solely chosen based on the current local observations (rather than the entire history of each agent's respective local observations).  Again, it is not clear to me that this assumption is intrinsically necessary, except for making the action-prediction problem of LIAM simpler.  Any reasons why more general history-based policies were not considered for the modeled agents?

5) The NAM baseline is described as being a "lower performance of LIAM", and I'm not convinced that this is strictly true.  While LIAM does seem like a useful task, there is no reason to believe that it is always strictly better than no auxiliary task at all (as in NAM).  Similarly, I am not convinced that FIAM is indicative "upper performance of LIAM".  Again, LIAM and FIAM are similar but ultimately simply different tasks, one more centered on the ability to predict other agents, and the other on the ability to synthesize the agent's own history.  While both are relevant for decision making, claiming that the information encoded by FIAM is strictly better than that encoded by LIAM needs backing.  I would assume that broadly speaking, LIAM, FIAM, and NAM are stronger in different types of domains.  Is there any further evidence for these statements, or am I maybe reading those statements as more strongly than the authors intended?

6) Why is the decoder architecture factored into the observation and policy reconstruction models, as opposed to a single model which outputs both observation and action?  (aside, perhaps, to facilitate the ablation study?).  Is the observation used in the policy reconstruction model always the one generated by the observation reconstruction model, or do you also use the ground truth observation, akin to teacher forcing?

The submission reads very well, and there are very few minor issues concerning readability:

* l.220:  "than" -> "then".

* l.270:  Confidence interval of what kind of uncertainty measure?  Standard deviation or standard error?  I assume standard deviation, but this should be stated explicitly.

* Fig.4. (and to a lesser extent, Fig.3):  Labels and ticks are too small and should be enlargened.

**Time Spent Reviewing:**

6

---

> ### Author Response · Authors · 2021-08-10
> **Response to reviewer 5w5T**
>
> We would like to thank reviewer 5w5T for their review.
>
> ### Main issues:
>
> 1. NAM (which is just how we refer to the RL$^2$ algorithm applied in our context) and VariBad receive the reward as input, while FIAM, LIAM, CBAM, CARL do not use the reward, as correctly pointed out by the reviewer. In the paper, we chose to use rewards in the input to NAM/RL$^2$ and VariBad in order to stay consistent with their original implementations. However, experimentally we found that in none of our considered environments the use of the rewards affects the performance of NAM and VariBad, and thus we left these results out. For example, reward in DSL: the reward depends on the distance of both agents from their correct landmark, and cannot be used to successfully guide the controlled agent to its landmark (even if the controlled agent moves to the correct landmark, it will still be heavily penalised if the modelled agent is not in the correct landmark). We will include these clarifications (model inputs) and additional results (using no reward in NAM/VariBad) in the paper/appendix.
>
>
> 2. The assumption that other agents use one set of fixed policies is common in the literature, e.g. [1,2,3,4]. LIAM has been developed around this experimental framework. We believe this assumption is a good starting point to tackle the relatively unexplored problem of representation learning for agent modelling under partial observability, where having all agents learn concurrently would substantially complicate the problem. We agree that extending this work to allow for non-stationarity (e.g. as exhibited in multi-agent RL) would be a very interesting direction for future work.
> Furthermore, we can use LIAM to train agents and then evaluate them against policies that are not seen during training. In DSL, we cannot expect the controlled agent to generalise to unseen policies, since different combinations of communication messages are out of the training distribution.
> We performed some extra experiments to evaluate the generalisation of LIAM against unseen fixed policies in LBF. We train LIAM against five of the fixed policies, while we use the other five to evaluate how well it generalises. We also performed the same experiments for NAM. (we report results for only one seed)
>
>     LIAM:  0.4090(train)    0.3859(gen)
>
>     NAM:   0.3795(train)    0.3610(gen)
>
>     We observe that the drop in generalisation returns is relatively small compared to the training returns.
>
>
> 3. We use two decoding networks in LIAM: the observation reconstruction and the policy reconstruction decoder. These two modules reconstruct the joint observation and action respectively of all modelled agents. The dimensions of the joint observation and action of the modelled agents grow linearly with respect to the number of agents. Therefore, the outputs' dimensions of the two decoding networks also grow linearly with respect to the number of modelled agents. This is a manageable dimension even for relatively large multi-agent systems. We will clarify the statement noted by the reviewer in the future revision of the paper.
>
> 4. LIAM can model agents independently from whether they choose their actions in a Markovian way or not. A recurrent decoder can be used if we consider that the modelled agent computes its actions based on the history of its observations.
> The assumption about the reactive agents is not as strict as it is described in Appendix D. In Appendix D, we describe an extreme case where neither the observations nor the actions of the modelled agent are changing during an episode. Therefore, the effect of the modelled agent in the controlled agent during an episode is constant, and it is not feasible (to the best of our knowledge) to learn a model using only the observation action trajectory. However, in the majority of the environments, the trajectory of the controlled agent will be altered in some way during the episode by the modelled agent.
>
> 5. We do not aim to imply that FIAM and NAM are strictly (in the mathematical sense) upper and lower baselines. We will update the paper to make it clear.
>
> 6. Different factorisations of the modelled agent's trajectory are possible. We chose our factorisation as a natural way to factorise a trajectory since the action of the modelled agent depends on its observation:
>
>     $ p(o^{-1}_t, a^{-1}_t | z_t) = p(a^{-1}_t|o^{-1}_t, z_t) p(o^{-1}_t|z_t) $
>
>     However, it is possible to use a single decoder to reconstruct both the observation and action of the modelled agent. In the policy reconstruction decoder, we use as input the ground-truth observation of the modelled agent. Using the reconstructed observation as input to the policy reconstruction decoder would lead to an accumulation in the error.
> We will add an extra experiment in our ablation study in Section 4.5 to evaluate LIAM when the reconstruction of the observation and the action of the modelled agent is done by a single network.
>
> ### Minor issues:
>
> We would like to thank the reviewer for pointing us to these issues. We will address them in the future revision of the paper.
>
> In L270, we compute the confidence interval using the standard error.
>
> ### References
>
> [1] Albrecht et al. Reasoning about Hypothetical Agent Behaviours and their Parameters. AAMAS 2017
>
> [2] Grover et al. Learning Policy Representations in Multiagent Systems. ICML 2018.
>
> [3] Tacchetti et al. Relational Forward Models for Multi-Agent Learning. ICLR 2019.
>
> [4] Zintgraf et al. Deep Interactive Bayesian Reinforcement Learning via Meta-Learning. AAMAS 2021.

---

> > ### Comment · Reviewer_5w5T · 2021-08-12
> > **Clarification**
> >
> > Re. 5: Could you explain what you're going to clarify, exactly?  If not in a mathematical sense, in what way are FIAM and NAM upper and lower baselines?

---

> > > ### Author Response · Authors · 2021-08-13
> > > **Clarification**
> > >
> > > Thank you for your response.
> > >
> > > In the paper, we will clarify that FIAM and NAM are not strict (in the mathematical sense) upper and lower baselines to the performance of NAM, but that we intuitively expect them to perform better/worse than LIAM in our environments since these three algorithms have access to different information.
> > > FIAM has access to privileged information (the modelled agent's trajectory) both during training and execution. FIAM's policy is conditioned both on the observations of the controlled agent, as well as the embeddings that are generated by the trajectory of the modelled agent. In contrast, LIAM only has access to the modelled agent's trajectory during training but not execution.
> > > NAM does not explicitly model the other agent in the environment. LIAM has access to privileged information during training (trajectory of modelled agent), while NAM does not have access to such information. Therefore, we again expect that LIAM will achieve higher returns compared to NAM.
> > > These expectations regarding performance comparisons are in fact confirmed by our results shown in Fig 3, which show LIAM "between" FIAM and NAM.

---

> > ### Comment · Reviewer_5w5T · 2021-08-16
> > **More Comments**
> >
> > Another couple comments to the authors:
> >
> > 1.  I strongly encourage the authors to include in the *main* documents,
> >     experiments which are fair, i.e., where each method has access to the same
> >     information.  Whether the original works by NAM and VariBad included
> >     rewards or not is fundamentally an implementation/representational detail,
> >     rather than a fundamental aspect of their methods, and is not a good enough
> >     motivation for presenting results which *may* easily be skewed and biased.
> >     Luckily for the authors, this should be easy to address since, as they
> >     claim, the results are in practice equivalent whether even if the rewards
> >     are not included (although being able to see this would have been nice).
> >
> >     I also disagree with the authors' statement that the rewards cannot be used
> >     to successfully guide the controlled agent to its landmark.  Just because
> >     the reward is an aggregate of the status/position of all agents, does not
> >     mean that an agent cannot use it to determine its own level of success
> >     without needing to communicate, e.g., if the reward is close to zero, then
> >     every agent will know that it has reached its goal landmark without the
> >     need to communicate.  Aggregate strategies also exist, where all agents
> >     stand still except for one agent which takes a step in a couple of
> >     directions to determine the direction of its own goal, again, without the
> >     need to communicate.  Again, this will not be an issue as long as the main
> >     paper focuses on results in which none of the methods use *observable*
> >     rewards.
> >
> > 4. If there is no assumptions about whether the teammates are reactive or not,
> >    then the notation in lines 123-124 should be updated to make the teammate
> >    policies act based on their entire history, rather than only their current
> >    observation.

---

> > > ### Author Response · Authors · 2021-08-18
> > > **Response to reviewer 5w5T**
> > >
> > > Thank you for the additional comments.
> > >
> > > 1. We reran NAM without receiving the reward as input in DSL and the average return at the end of the training is: -16.8. This is similar to the returns of NAM that is presented in Figure 3 at the end of the training. We also want to emphasise that in the future revision of the paper, we will substitute the returns for NAM, and VariBad (that are currently presented in the paper), with the returns of NAM and VariBad without using the reward as input for all environments.
> > >
> > > 2. In our experiments, the fixed policies are not conditioned on the history of observations. However, application to fixed policies that are conditioned on the history of observations is possible by using a recurrent decoder. Thank you for pointing us to this issue. We will address that in the future revision of the paper.

---

### Decision · Program_Chairs · 2021-09-27

**Decision:**

Accept (Poster)

**Comment:**

This paper considers the problem of partial observability in multi-agent learning, and takes the approach of learning the latent representations of other agents from local observations. This is shown to be beneficial during learning.
The reviewers were generally positive about this paper, which presents an interesting and effective solution to an important problem.
The reviewer scores were improved as a result of the rebuttals, and the authors should include the promised revisions in the paper.